# A Review of Fault Diagnosis Methods for Rotating Machinery Using Infrared Thermography

**DOI:** 10.3390/mi13101644

**Published:** 2022-09-30

**Authors:** Rongcai Wang, Xianbiao Zhan, Huajun Bai, Enzhi Dong, Zhonghua Cheng, Xisheng Jia

**Affiliations:** Shijiazhuang Campus of Army Engineering University of PLA, Shijiazhuang 050003, China

**Keywords:** infrared thermography, infrared thermal image, rotating machinery, fault diagnosis

## Abstract

At present, rotating machinery is widely used in all walks of life and has become the key equipment in many production processes. It is of great significance to strengthen the condition monitoring of rotating machinery, timely diagnose and eliminate faults to ensure the safe and efficient operation of rotating machinery and improve the economic benefits of enterprises. When the state of a rotating machine deteriorates, the thermal energy that is much more than its normal operation will be generated due to the increase in the friction between the components or other factors. Therefore, using the infrared thermal camera to collect the infrared thermal images of rotating machinery and judge the health status of rotating machinery by observing the temperature distribution in the thermal images is often more rapid and effective than other technologies. Nevertheless, after decades of development, the research achievements of infrared thermography (IRT) and its application in various industrial fields are numerous and complex, and there is a lack of systematic sorting and summary of the achievements in this field. Accordingly, this paper summarizes the development and application of IRT as a non-contact and non-invasive tool for equipment condition monitoring and fault diagnosis, and introduces the basic theory of IRT, image processing technology and fault diagnosis methods of rotating machinery in detail. Finally, the review is summarized and some future potential topics are proposed, which will make the subject easier for beginners and non-experts to understand.

## 1. Introduction

Rotating machinery mainly refers to the mechanical equipment that can realize specific functions through rotating operation. It is widely used in petrochemical, electric power, metallurgy, aerospace and automobile manufacturing industries. Steam turbine, motor, fan, gearbox, reducer and engine are typical rotating machinery. With the continuous improvement of modern industrial production requirements for rotating machinery speed, load and automation, when rotating machinery fails, it is easy to cause equipment shutdown, leading to significant economic losses and even casualties. According to statistics, in industrial production, major accidents and economic losses caused by mechanical equipment failures account for about 38%, of which rotating machinery failures account for the majority [1]. It can be seen that real-time condition monitoring and fault diagnosis of rotating machinery, scientific and effective formulation of the maintenance strategies, ensuring the normal operation of rotating machinery and avoiding serious accidents are practical problems that need to be solved urgently in relevant industries. Fault diagnosis of rotating machinery has become a very important part in system design and maintenance [2], which is of great significance to improve economic benefits.

To improve the service life and safety of rotating machinery and reduce the shutdown loss, experts and scholars in related fields have studied and proposed a series of technical methods to diagnose the faults of rotating machinery. Common fault diagnosis methods for rotating machinery include fault diagnosis based on vibration signals [3,4,5,6], fault diagnosis based on acoustic emission technology [7,8], fault diagnosis based on temperature information [9,10], fault diagnosis based on oil analysis technology [11,12,13,14] and other non-destructive testing technologies [15]. For electric driven rotating machinery, fault diagnosis can also be carried out by collecting and analyzing the current and voltage signals [16,17]. The above technical methods have been fully studied in various fields, and have achieved a high accuracy in rotating machinery fault diagnosis. However, there are certain applicable conditions and shortcomings.

The fault diagnosis technology based on vibration signals has been relatively mature, and a complete set of intelligent state monitoring and evaluation systems have been basically formed [18,19]. The disadvantages mainly lie in the acquisition process of vibration signals. Vibration sensors need to be directly installed on the surface of rotating machinery, which leads to problems in the installation and configuration of sensors and even damage to rotating machinery. In addition, when rotating machinery is in operation, a large number of noise signals will be generated in the process of vibration signal acquisition, which is not conducive to the subsequent signal processing and fault diagnosis. Acoustic emission technology is a real-time dynamic monitoring technology with a simple and clear operation process. It has been widely used in the condition monitoring and fault diagnosis of mechanical equipment. The disadvantages are that there are many characteristic parameters of acoustic emission and there is no unified standard for the representation of signal characteristics, which is not conducive for improving the detection accuracy. The oil analysis technology uses the equipment working condition information contained in the oil to judge the current and future states of the equipment, thus providing an effective basis for the correct maintenance of the equipment. At the same time, there are some limitations in the application of oil analysis technology, that is, the fault standard of equipment is not easy to define, and if the fault of rotating machinery is not caused by wear, it is often difficult to judge its health status.

Temperature information is also an important signal reflecting the health status of mechanical equipment [20]. The use of temperature change for fault diagnosis of rotating machinery has been paid attention by relevant researchers, and the application is simple and effective. Usually, the thermocouple or resistance temperature detector is used to obtain the temperature change in the equipment and detect the fault related to temperature abnormality in advance, so as to carry out preventive maintenance before the fault occurs. However, various temperature measurement systems commonly used at present are generally contact types. Temperature information detection is easily affected by equipment materials and environmental factors, and cannot form a visual image of the overall temperature distribution of the equipment, and the equipment under testing has certain limitations.

For the rotating machinery in operation, it is usually difficult to arrange corresponding sensors on its surface due to the rotating action of its components, and it is impossible to directly obtain the state information of the rotating machinery. When rotating machinery breaks down, conditions different from those of normal operating equipment usually occur, such as abnormal vibration and temperature change [21]. Therefore, infrared thermography (IRT) has gradually become a new non-destructive testing technology for measuring equipment temperature changes in recent years. This method has the advantages of non-contact, no shutdown, high monitoring efficiency and simple operation [22], and can provide infrared thermal images of the whole equipment or key components to indirectly obtain the visual state information of the equipment. At present, infrared thermal imaging has been successfully applied to condition monitoring and fault diagnosis of equipment in various fields, such as civil structure detection [23,24,25,26,27], material fatigue damage assessment [28,29,30], electrical equipment detection [31,32,33,34], tensile and plastic deformation detection [35,36], chemical vapor deposition process assessment [37,38], mechanical equipment fault diagnosis [39,40,41], etc., as shown in Figure 1.

At present, great progress has been made in using IRT for fault diagnosis of rotating machinery in industrial production. Although, compared with other fault diagnosis methods of rotating machinery, the fault diagnosis method based on IRT is not mature, the relevant standards and mechanisms are not perfect and the fault diagnosis process is not fully intelligent. At home, infrared temperature measurement is only used as an auxiliary means to monitor the temperature change in rotating equipment, and foreign countries are gradually exploring the use of IRT for quantitative and systematic analyses of rotating equipment. However, as far as the rotating machinery is concerned, when its state deteriorates, the thermal energy between the components will be generated much more than its normal operation due to the increase in friction or other factors. Therefore, using the infrared thermal camera to collect the infrared thermal images of rotating machinery and judge the health status of rotating machinery by observing the temperature distribution in the thermal images is often more rapid and effective than other technologies.

At the theoretical research level, professor Younus and Yang [42] of Pukyong University in South Korea conducted a relatively full study on the specific methods and processes of fault diagnosis for mechanical equipment by using an infrared thermal camera. Meola [43] introduced the origin of IRT, the development of infrared detectors and the application prospects of IRT in detail in his work. The specific flow of fault diagnosis methods for rotating machinery based on IRT is infrared thermal image acquisition, image preprocessing, feature parameter extraction and fault diagnosis. In this paper, we sorted out and summarized the principle and technical background of IRT, built a hardware platform for IRT applications and comprehensively analyzed the flow of fault diagnosis methods for rotating machinery and the technical methods involved, which will make the subject easier for beginners and non-experts to understand, more conducive to the development of IRT. The basic principles and experimental methods of IRT will be combined to introduce each step of fault diagnosis methods for rotating machinery in detail.

## 2. The Principle of Infrared Thermography

### 2.1. The Background of Infrared Thermography

The International Organization for Standardization (ISO) defines the IRT technology as a technology that collects and analyzes the heat distribution information on the surface of an object through a non-contact infrared imaging device [44]. In 1800, infrared light was first discovered by British astronomer Herschel. The origin and basic theory of IRT are described in detail in the literature [45]. Matter in nature, as long as the temperature is higher than absolute zero (0 K or −293 °C), will radiate electromagnetic waves to the outside world. This radiation is mainly related to the temperature and material of the material, so it is also called thermal radiation. The spectrum of thermal radiation is a continuum, and theoretically, the wavelength can be from 0 to ∞, and the general thermal radiation is mainly propagated by the infrared rays in invisible lights, so thermal radiation is also called infrared radiation. Usually, according to the wavelength or frequency of the electromagnetic wave, it can be divided into different bands, namely the electromagnetic spectrum, as shown in Figure 2.

As can be seen from Figure 2, infrared belongs to invisible light and is located between visible light and microwave in the whole electromagnetic spectrum, with a wavelength range of 760 nm to 1 mm. In the electromagnetic spectrum, the wavelength and frequency of the electromagnetic waves are inversely proportional, and the visible light only occupies a small part of the entire electromagnetic spectrum. In the infrared band, it can be subdivided into near-infrared (0.76∼3 μm), mid-infrared (3∼15 μm) and far-infrared (15∼1000 μm). The atmosphere and smoke clouds absorb the visible light and near-infrared rays, but they are relatively transparent to the infrared rays of 2∼2.5 μm, 3∼5 μm and 8∼14 μm. Therefore, these three bands are called the “atmospheric window” of infrared rays. In the atmospheric window, the amount of infrared rays absorbed by the atmosphere and smoke clouds is small. The temperature measurement and imaging function of the infrared thermal camera are realized by detecting the infrared radiation of the object in the atmospheric window and according to the basic law of the infrared radiation.

In fact, the thermal radiation process of an object is very complicated. The blackbody model proposed by German physicist Kirchhoff is the basis for studying the theory of thermal radiation of objects and the application of infrared technology. The so-called blackbody refers to an object that can completely absorb the electromagnetic radiation of all wavelengths without reflection. At the same time, the blackbody is also a complete radiator and can radiate electromagnetic waves of various wavelengths to the outside world. The blackbody model is an ideal model, which shows that there is a quantitative relationship between infrared radiation, wavelength and temperature.

On this basis, the German theoretical physicist Planck proposed the famous Planck’s law of blackbody radiation (Planck’s law for short) according to the blackbody model and quantum theory; that is, the blackbody can radiate a continuous spectrum in the process of thermal radiation:(1)$$E_{b\lambda }=\frac{c_{1}}{\lambda^{5}[\exp(\frac{c_{2}}{\lambda T})-1]}$$
where λ is the wavelength of the electromagnetic radiation, and the common unit is μm; $E_{b\lambda }$ represents the spectral radiant emissivity, which refers to the amount of energy radiated by a radiation source in unit area and unit time for a specific wavelength. The common unit of $E_{b\lambda }$ is $W\cdot cm^{-2}\times \mu m^{-1}$; T is the thermodynamic temperature, whose unit is K; $c_{1}$ and $c_{2}$ are the first radiation constant and second radiation constant, respectively. Planck’s law explains the law of radiation energy distribution of the blackbody, according to wavelengths, and gives the radiation spectrum distribution characteristics of the blackbody at different temperatures, as shown in Figure 3.

Figure 3 shows that when the temperature is determined, the spectral radiant emissivity will first increase and then decrease with the increase in wavelengths; that is, the spectral radiant emissivity change curve has a maximum value. When the wavelength is determined, the spectral radiant emissivity of the blackbody will increase with the increase in temperatures. At different temperatures, the peak spectral radiant emissivity of the electromagnetic waves has a linear relationship with the wavelength, and with the increase in the wavelength, the peak spectral radiant emissivity gradually decreases.

Planck’s law provides a theoretical basis for the research and application of IRT technology. Before Planck’s law was formally put forward, Slovenian physicist Stefan proposed the blackbody radiation integral law by summarizing the experimental data. After that, the Austrian physicist Boltzmann, based on the thermodynamic theory, assumed that electromagnetic wave radiation was used instead of gas as the working medium of the heat engine and finally derived the same conclusion as the induction result of Stefan. Finally, it was collectively referred to as Stefan–Boltzmann law.

The Stefan–Boltzmann law is only applicable to the ideal radiation sources, such as blackbody, which describes the law of the total radiance of all wavelengths emitted by the whole hemispherical space on the surface of the blackbody changing with temperatures. It can be obtained by integrating all wavelengths in Planck’s law. The mathematical expression is as follows:(2)$$J(T)=\varepsilon \sigma T^{4}$$

Among them, the scale coefficient σ is the Stefan–Boltzmann constant, which can be calculated from other known basic physical constants in nature, and its value is about $5.67\times 10^{-8}\;W/(m^{2}\times K^{4})$; ε is the surface emissivity of the blackbody at a certain wavelength λ and absolute temperature T. If it is an absolute blackbody, then ε=1; J(T) is the total radiance of various wavelengths emitted per unit area in the whole hemispherical space of the blackbody surface. This law indicates that as long as the temperature of an object is higher than absolute zero (0 K), it will radiate heat to the outside, and the total energy J(T) radiated per unit area of the blackbody surface in unit time is proportional to the fourth power of the thermodynamic temperature T of the blackbody itself. Thus, it can be directly explained that even if the temperature of the object changes only a little, the total radiance of the object will change greatly, which provides a theoretical basis for improving the measurement accuracy of the IRT technology.

After Planck’s law and the Stefan–Boltzmann law, German physicist Wien proposed Wien’s displacement law, proving that the peak wavelength of the blackbody spectral radiance is inversely proportional to the absolute temperature of the blackbody surface:(3)$$\lambda_{\max}T=2897.7\;\mu \mathrm{mK}$$
where $\lambda_{\max}$ is the peak wavelength of the radiation, and T is the absolute temperature of the blackbody (unit: K). Wien’s displacement law states that the higher the temperature of an object, the shorter the wavelength of its radiation spectrum or the higher the frequency of its radiation spectrum. Table 1 shows the peak wavelengths of the emission spectra corresponding to relevant physical events at different absolute temperatures [15]. 

Table 1 indicates that the highest achievable thermal radiation is included in the infrared region. By analyzing Planck’s law, the Stefan–Boltzmann law and Wien’s displacement law, it can be seen that as long as the temperature of an object is higher than absolute zero, it will always radiate infrared rays to the outside world. This radiation is mainly determined by the temperature of the material. For objects of the same material, when the environment is determined, the energy radiated outward is only related to the surface temperature, and the higher the temperature, the greater the energy radiated outward, which is reflected in the infrared thermal image, that is, the higher the brightness. To sum up, Planck’s law, the Stefan–Boltzmann law and Wien’s displacement law reveal the basic principle of IRT and lay a solid theoretical foundation for the application of IRT in the field of fault diagnosis for rotating machinery.

### 2.2. The Imaging Principle of Infrared Thermal Camera

IRT is a technology that uses an infrared thermal camera to detect the infrared radiation emitted by objects in a non-contact manner. At the same time, after detecting the radiation power of rotating machinery, the surface temperature of the object can be calculated in real time according to the Stefan–Boltzmann law. Because the temperature distribution on the surface of rotating machinery is different during operation, the radiation intensity on the surface is different. The thermal images collected by the infrared thermal camera use different colors and brightness to represent different radiation intensities, so different radiation intensities correspond to different surface temperatures. As shown in Figure 4, the infrared thermal image of a reducer test bench can be used to monitor the surface temperature of the whole reducer. If the reducer is about to fail or has failed, its health states can be judged according to the color distribution of the infrared thermal image or the abnormal temperature of a pixel.

It can be seen from Figure 4, the infrared thermal image intuitively shows the temperature field distribution on the surface of equipment with different colors, which contains rich health information of the equipment. Infrared thermal cameras are the core of IRT systems. With the rapid development of modern science and technology, the structure and function of the infrared thermal camera become more complex and intelligent, and has been gradually applied to various fields of production and life [46]. Infrared thermal cameras are usually composed of an optical device, infrared detector, video amplifier and display. Infrared radiation carries the characteristic information of the object, which provides an objective basis for using the IRT technology to distinguish the temperature and thermal distribution of various measured objects. Using this characteristic, after the power signals radiated from the heating part of the object are converted into electrical signals by the optical device and the infrared detector, the video amplifier can correspondingly simulate the spatial distribution of the temperature on the surface of the object. Finally, the thermal image video signal is formed by the system processing and transmitted to the display to obtain the thermal image corresponding to the temperature distribution on the surface of the object, that is, the infrared thermal image. The structure composition and imaging principle of the infrared thermal camera are shown in Figure 5.

The main advantage of fault diagnosis and condition monitoring technologies based on IRT is that it requires less instruments. The basic hardware requirements of this application are an infrared thermal camera, a tripod and a video output unit for displaying the acquired infrared thermal images. Figure 6 is a schematic diagram of a typical IRT experimental device, which includes the infrared thermal camera, display, infrared thermal images of the gear reducer and its original photos.

The equipment and information contained in Figure 6 constitute the basic health monitoring system of rotating machinery based on IRT. In practical application, basically all rotating machinery will have temperature changes to a certain extent in case of failures or abnormal states, and IRT can simply and effectively monitor its real-time states, timely detect faults and eliminate potential faults, and prevent major accidents or economic losses.

## 3. The Flow of Fault Diagnosis Methods for Rotating Machinery Using Infrared Thermography

The infrared thermal camera is easy to be affected by environmental factors when working. Some important parameters will directly affect the clarity and accuracy of the infrared thermal images. Therefore, when using IRT to diagnose the fault of rotating machinery, it is first necessary to set the parameters of the infrared thermal camera. Some of the literature [47,48] has carried out detailed research on several important parameters that affect the working performance of the infrared thermal camera, such as the temperature range, spectral range, temperature resolution and spatial resolution. In the process of engineering applications and experiments, it is particularly important to reasonably select the infrared thermal camera. In addition to considering the influence of the above parameters, it also needs to be analyzed according to the actual situation of mechanical equipment. An infrared thermal camera is used to collect the infrared thermal images during the operation of rotating machinery, which is used as the original data for subsequent image processing, feature parameter extraction and fault diagnosis. The specific process is shown in Figure 7.

Figure 7 shows that the fault diagnosis of rotating machinery based on IRT is a logical process, and steps such as infrared image preprocessing and feature parameter extraction will have an important impact on the results and accuracy of the fault diagnosis.

### 3.1. Infrared Thermal Image Preprocessing

The essential step of fault diagnosis methods for rotating machinery based on IRT is to preprocess the collected infrared thermal images. The images of nature appear in the form of simulation. Only after the simulated images are digitized can the computer flexibly process the images. In the IRT system, to process the images, there is a step of simulated signals quantization into digital signals, so as to obtain the final infrared digital images. Therefore, infrared thermal image preprocessing belongs to the category of digital image processing. 

The basic contents and main methods of digital image processing are introduced in detail in the literature [49]. Digital image processing is a theory, method and technology for noise reduction, transformation, restoration, segmentation, enhancement and feature extraction of images by the computer, which belongs to the subclass of signal processing. Combined with the practical application of IRT, this paper focuses on image denoising, segmentation and enhancement.

#### 3.1.1. Image Denoising

In the processes of image acquisition, transmission and storage, it is often disturbed and affected by various noises, which makes the image quality decline. To obtain a high-quality digital image, it is necessary to eliminate the noise of the image and keep the integrity of the original information as much as possible. In the process of acquiring infrared thermal images affected by space–time environment, detection devices, circuit conditions, channel transmission errors and other factors, infrared thermal images often contain more complex and strong noises than ordinary optical images. From the visual point of view, the noise signals in the infrared thermal image are mostly isolated pixel points and blocks, and have no correlation with the surrounding pixels, so it will produce a relatively obvious contrast effect. The noise in the infrared thermal image will have a serious impact on the subsequent image processing, such as image segmentation, image enhancement, pattern recognition, etc., and even cover or lose the information originally expressed by the image. Therefore, infrared thermal image denoising has an important research significance.

At present, relevant scholars at home and abroad have proposed a variety of noise reduction algorithms for digital image noises and achieved good results [50,51,52,53], many of which can be directly applied to the noise reduction process of infrared thermal images. At present, the noise reduction methods of digital images can generally be divided into two categories: traditional noise reduction methods and deep-learning-based noise reduction methods [54].

The traditional image denoising methods are usually divided into spatial domain method, transform domain method and a combination of the two methods. The spatial method is to directly calculate the grayscale values corresponding to each pixel in the digital image. The noise reduction algorithm based on transform domain is an indirect noise reduction algorithm, which modifies the transform coefficient value of an image in a certain transform domain. Mean filter [55,56,57], median filter [58,59] and Wiener filter [60,61] are classical algorithms for spatial noise reduction. However, the noise reduction principle of the above three image denoising algorithms is to adopt the same noise reduction model for the edge region and the inner region in the image, which will cause the loss of texture feature information while filtering out noises [62]. The commonly used transform domain methods mainly include Discrete Fourier Transform (DFT) [63], Fast Fourier Transform (FFT) [64], Discrete Cosine Transform (DCT) [65], etc. Different transform domain methods will make each region of the image show different characteristics. After the DCT, the inner region of the image is mostly low-frequency components, while the image noises and the edge regions are realized as high-frequency components. Therefore, low-pass filtering can be used to achieve the purpose of noise reduction, but the edge texture of the image will be lost to a certain extent. With the gradual maturity of image denoising methods based on spatial domain and transform domain, many scholars began to combine the two, providing a new idea for image denoising. 

The proposal and development of partial-differential-equation-based algorithms [66], non-local average value algorithms [67], three-dimensional block matching algorithms [68], weighted kernel norm minimization algorithms [69] and other methods have achieved good results in image denoising. At present, the improvement of traditional image denoising methods is mostly based on three-dimensional block matching algorithms and weighted kernel norm minimization algorithms. It can be seen that the traditional image denoising methods are mainly aimed at certain noise models, such as Gaussian white noises, pepper and salt noises, and do not take into account other noises with unknown statistical laws contained in infrared thermal images. Therefore, the traditional image denoising methods only aim at some specific noises, and there are some limitations. For infrared thermal images with mixed noises, the denoising effect is not so good.

In recent years, with the rapid development of the deep learning technology, many scholars have applied it to image processing, pattern recognition and other fields, improving the effect of image denoising. Existing noise reduction methods based on deep learning can be divided into two types: image denoising methods based on multi-layer perceptrons (MLP) [70] and image denoising methods based on convolutional neural networks (CNN). MLP has a good nonlinear learning ability, but the number of parameters that need to be adjusted in the process of model training is very large. The noise reduction methods based on CNN can effectively reduce the parameters of model training, further extract the deep-seated features of the image and reduce the occurrence of overfitting [71,72,73]. Among them, the more representative noise reduction methods based on CNN include encoder–decoder networks [74,75], nonlinear reaction–diffusion model [76], denoising convolutional neural networks (DNCNN) [77,78], etc. Using the idea of residual networks (ResNet) [79,80] for reference, DNCNN has achieved remarkable results in digital image denoising, greatly improving the training speed and accuracy of the model.

Compared with the traditional image denoising methods, the deep-learning-based denoising methods have made great improvements in noise filtering and detail retention, but there are also some limitations. With the deepening of the neural networks and the improvement of the complexity, the training difficulty of the model is greatly increased, even the gradient disappears, and the noise reduction effect of the image is difficult to further improve. Therefore, it has a very important theoretical research significance and application value to realize image denoising and detail information retention and reconstruction simultaneously.

#### 3.1.2. Image Segmentation

In the research and application of images, people are often only interested in some objects in the image, which usually correspond to the areas with specific properties in the image. Image segmentation refers to the technology and process of dividing an image into regions with corresponding characteristics and extracting the objects of interest. These characteristics can be grayscale, color, texture, etc. The object of interest can correspond to a single region in the image or multiple regions.

The area after image segmentation shall have the following characteristics:

(1) The segmented regions are consistent in some features (such as grayscale, color, texture, etc.);

(2) the area is single inside without too many small holes;

(3) the adjacent regions have obvious differences in the characteristics on which the segmentation is based;

(4) the division boundary is clear.

To analyze and evaluate the health status of rotating machinery more accurately, it is necessary to separate and extract the target area from the infrared thermal image, and then further study and analyze it. In academic research and practical applications, image segmentation methods can generally be divided into two categories. One category is traditional image segmentation methods, mainly including threshold segmentation [81], edge segmentation [82,83], region segmentation [84,85], clustering segmentation [86], mathematical morphology segmentation [87], etc. The special point is that image features need to be manually selected. The other is image segmentation based on CNN, which can automatically extract image features. On the basis of CNN framework, Long et al. [88] proposed full convolution networks (FCN), which further improved the accuracy and generalization ability of CNN in image segmentation. In the follow-up research process, SegNet [89] and U-Net [90] were successively proposed based on the design idea of FCN framework. The ideas of SegNet and FCN are very similar, but SegNet proposes an encoder–decoder CNN, which can improve the edge characterization and reduce the training parameters; U-Net is a semantic segmentation network based on FCN, which is widely used in medical image segmentation. Based on image segmentation, feature extraction and parameter measurement are carried out for infrared thermal images, which makes it possible for higher-level image analysis and understanding. Therefore, the research on image segmentation methods is of great significance.

#### 3.1.3. Image Enhancement

Image enhancement is a common image processing technology in infrared thermal image preprocessing. It is mainly to improve the quality of the image and enhance the part of interest. By emphasizing or sharpening some features of the image, such as edge, contour, contrast, etc., the visual effect of the image can be improved or the image can become more convenient for computer processing. For example, the image with low light needs to be enhanced in brightness.

In the process of infrared thermal image acquisition, the image may be affected by the space–time environment, detection devices, circuit conditions, channel transmission errors and other factors, resulting in blurred details and low contrast. Image enhancement technology can effectively solve the above problems. In essence, after the infrared thermal image is enhanced, the amount of the characteristic information represented by the infrared thermal image does not increase or decrease, only the dynamic change range of the characteristic information is changed, thus making it easier for the computer to detect and recognize the characteristic information of the image.

The image enhancement technology can also be divided into two methods in spatial and frequency domain, mainly including histogram equalization [91,92,93,94], grayscale transformation [95], fuzzy technology [96,97], image smoothing [98,99], image sharpening [100,101] and two-dimensional empirical mode decomposition [102,103]. At present, histogram equalization is one of the most commonly used methods in the field of image enhancement. Many pieces of the literature have introduced this method and its improved forms. The grayscale histogram of an image describes the grayscale distribution in the image and can intuitively show the proportion of each grayscale level in the image. Its main idea is to change the histogram distribution of a pair of images into an approximate uniform distribution through the cumulative distribution function to enhance the contrast of the image [104]. However, since histogram equalization significantly changes the contrast of the image, it is easy to cause the loss of detail information [105]. In view of this situation, scholars at home and abroad have proposed many improved algorithms, such as adaptive histogram equalization [106,107,108] and unsharp mask method [109], which retain the details of the image and improve the effect of image enhancement. By enhancing the infrared thermal image, it is more advantageous to extract the feature parameters in the next step.

### 3.2. Feature Parameter Extraction

Compared with the traditional fault diagnosis technology for rotating machinery, the state information of rotating machinery can be easily obtained by using IRT. To obtain equipment state information from infrared thermal images, it is necessary to apply a variety of image processing technologies such as DFT, DCT, wavelet transform (WT) and neural networks [42]. After the infrared thermal image is enhanced, the difference between the images corresponding to different health conditions of rotating machinery can be observed in some cases, but it is still necessary to extract the characteristic parameters of the image for quantitative analysis. Feature parameter extraction is a key step in fault diagnosis of rotating machinery. Extracting feature parameters that are sensitive to the change in rotating machinery health condition is very important for subsequent fault pattern recognition. Image features mainly include histogram values, spectrum, texture, shape and color [110]. Histogram values are mainly image feature parameters obtained from the perspective of statistics, which can comprehensively represent image information and characteristics, including mean, variance, inclination, kurtosis, energy, entropy, etc.

However, because the manually extracted infrared image feature parameters cannot fully characterize the fault state of rotating machinery, the accuracy of fault diagnosis is not high, and the research of the related literature also have certain limitations. In recent years, with the birth of large-scale image data sets and the rapid development of computer software and hardware, CNN has been widely used in computer vision and speech recognition and achieved good results. Compared with the traditional feature parameter extraction methods, CNN can automatically extract the feature information of images, effectively learn the deep-seated features from a large number of samples, avoid the complex feature parameter extraction process and require only a small amount of manual participation in the whole process. Krizhevsky et al. adopted the AlexNet network and won the championship of ImageNet 2012 image recognition challenge with great advantages, marking the successful application of CNN in large-scale image recognition. Subsequent researchers have further improved the structures of neural networks on this basis, such as GoogleNet, ResNet, DenseNet [111], etc., which can more effectively carry out feature extraction and target recognition. The research process [31] of some representative feature extraction networks and target detection models is shown in Figure 8, where the upper and lower parts of the coordinate axis, respectively, represent the target detection model and the feature extraction network.

As shown in Figure 8, in the recent ten years, the target detection model and feature extraction networks have been fully developed, especially with the deepening of the research of CNN, which greatly improves the accuracy of target detection and the efficiency of feature extraction.

### 3.3. Fault Diagnosis

Fault diagnosis is a process of information output on the basis of data collection and processing in the early stage. It makes a judgment on the fault symptom, position and type of fault and degree of fault. Because the fault modes of rotating machinery are generally complex, to improve the reliability of rotating machinery and meet the growing demands for fault detection, the fault diagnosis methods of rotating machinery have been fully developed and applied in recent years.

At present, the commonly used fault diagnosis methods can be divided into qualitative analysis methods and quantitative analysis methods, and various fault diagnosis methods can be compared and analyzed from the advantages and disadvantages, as shown in Table 2.

Through the interpretation of Table 2, it can be seen that there are many methods of fault diagnosis for rotating machinery, and different methods have their own adaptability and limitations. In recent years, with the rapid development of computer technology, the fault diagnosis method based on machine learning has attracted more and more attention from the academic and industrial circles and has achieved a high diagnostic accuracy in practical applications. Machine learning algorithms can be divided into two categories. One is the traditional machine learning algorithm [129,130,131,132], including support vector machine (SVM), relevance vector machine (RVM), k-nearest neighbor (KNN), artificial neural networks (ANN), decision tree, Bayesian classifier, etc. Taheri-Garavand et al. [133] took advantage of two-dimensional discrete wavelet transform (DWT) to extract image features, and proposed an intelligent fault diagnosis system for radiators under different working conditions based on infrared thermal images. Finally, ANN was used for classification. Tran et al. [134] used two-dimensional empirical mode decomposition (EMD) for infrared image enhancement and used RVM for fault diagnosis. The effectiveness of this method was verified by comparison with the classification results of SVM and adaptive neuro-fuzzy inference system. The other is the deep learning algorithm [135,136,137,138], which is also the frontier direction in the field of machine learning at present. It mainly includes CNN, multi-layer perceptrons (MLP), deep belief networks (DBN), recurrent neural networks (RNN), generative adversarial networks (GAN), etc. The traditional machine learning algorithm needs to manually extract the characteristic parameters of the images when classifying the infrared thermal images, and its self-learning and self-adaptive abilities are poor. In contrast, as one of the most commonly used methods in the field of image recognition, deep learning algorithms such as CNN have a good self-learning ability and generalization ability, and can omit the complex feature extraction process before image recognition. At present, the fault diagnosis method based on deep learning has become a research hotspot, and many researchers have also conducted in-depth research in this field. Li et al. [24,139] made use of IRT to deeply study the fault modes of rotor systems and industrial gearbox systems, and the multi-scale CNN was used to classify the fault modes, which achieved good results. Wang et al. [140] proposed a crack recognition method based on IRT to solve the problems of low efficiency and poor anti-interference ability of traditional non-destructive testing technologies in steel plate crack detection. CNN was used for image recognition and classification, and high recognition accuracy was achieved in the test set. Choudhary et al. [141] collected the infrared thermal images of the rolling bearing under six different states and used the LeNet-5 structure of ANN and CNN to classify the images. By comparing and analyzing the classification results of the two methods, the superiority of CNN is proved.

## 4. Conclusions and Future Works

Rotating machinery fault diagnosis is an important link in modern industrial production, which can effectively prevent the occurrence of catastrophic accidents and reduce economic losses. After the above analysis, the following conclusions can be drawn:(1)At present, IRT has developed into an effective tool for the fault diagnosis of rotating machinery. It can carry out real-time non-contact temperature monitoring of the equipment, provide information about the equipment health status and production process efficiency and play a vital role in rotating machinery health status monitoring. With the continuous development and progress of science and technology, the accuracy and work efficiency of the infrared thermal camera will be further improved, thereby promoting the application of IRT in various fields.(2)In this paper, the related research status of the fault diagnosis methods for rotating machinery is summarized, and the flow of fault diagnosis methods for rotating machinery based on IRT is proposed. The application of modern image processing technologies and deep learning methods in infrared thermal image processing and classification can speed up the decision-making process, improve the accuracy of image classification and avoid the interference of human factors. Especially for mechanical equipment, the infrared thermal image may not be clear due to its thick metal casing. At this time, more efficient deep learning algorithms are needed for image preprocessing to better realize the fault diagnosis of mechanical equipment.(3)IRT has profound principles, which will have an important impact on the improvement and application of the IRT technology. This research will help non-professionals in this field and practitioners in related industries to apply this technology in the condition monitoring of rotating machinery, to reduce equipment downtime and maintenance costs and improve production efficiency.

This paper demonstrates several opportunities for future work.

(1)Because the deep learning model requires sufficient samples during training, which consumes a lot of computer resources, the study of image compression sensing technologies is helpful to improve the diagnostic efficiency of infrared thermal images for rotating machinery.(2)As a typical equipment of the industrial machinery, rotating machinery has a complicated failure mechanism. Therefore, it is necessary to further improve the failure mechanism analysis when using IRT to monitor the state of rotating machinery.(3)On this basis, it is of great significance to deepen the application of CNN and other deep learning algorithms, establish a rotating machinery condition monitoring and fault diagnosis system and improve the accuracy and generalization ability of the model.

## Figures and Tables

**Figure 1 micromachines-13-01644-f001:**
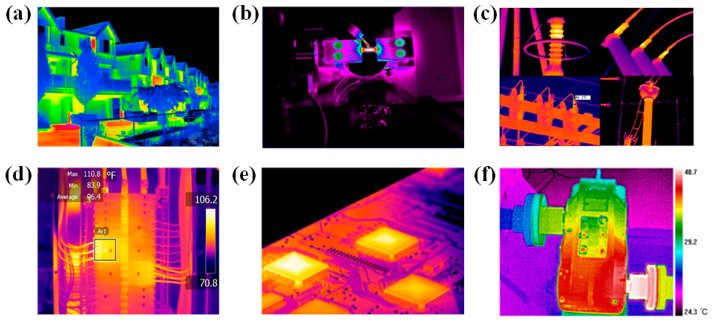
Some applications of infrared thermography technology: (**a**) Inspection of exterior structure. (**b**) Material fatigue damage assessment (**c**) Health status monitoring of the electrical equipment. (**d**) Inspection of corroded parts of oil refining equipment (**e**) Defect detection of circuit board. (**f**) Fault diagnosis of the mechanical equipment.

**Figure 2 micromachines-13-01644-f002:**
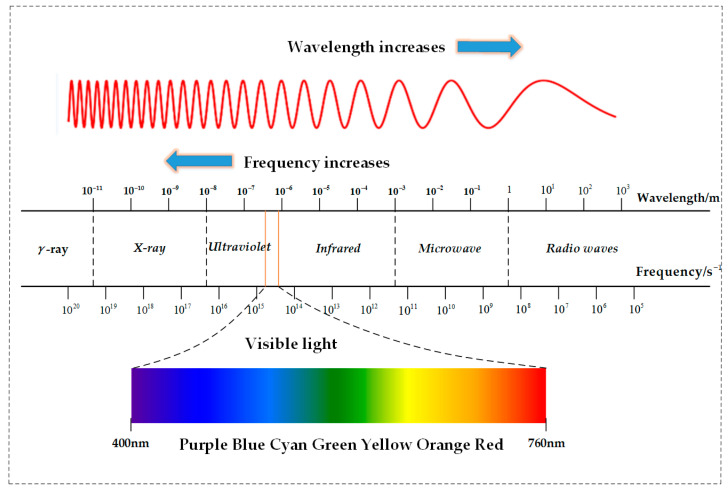
The electromagnetic spectrum.

**Figure 3 micromachines-13-01644-f003:**
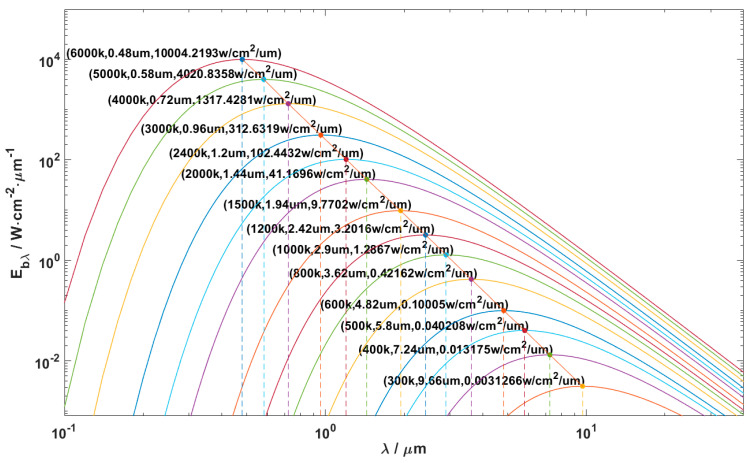
The variation rule of the blackbody spectral radiant emissivity with wavelengths at different temperatures.

**Figure 4 micromachines-13-01644-f004:**
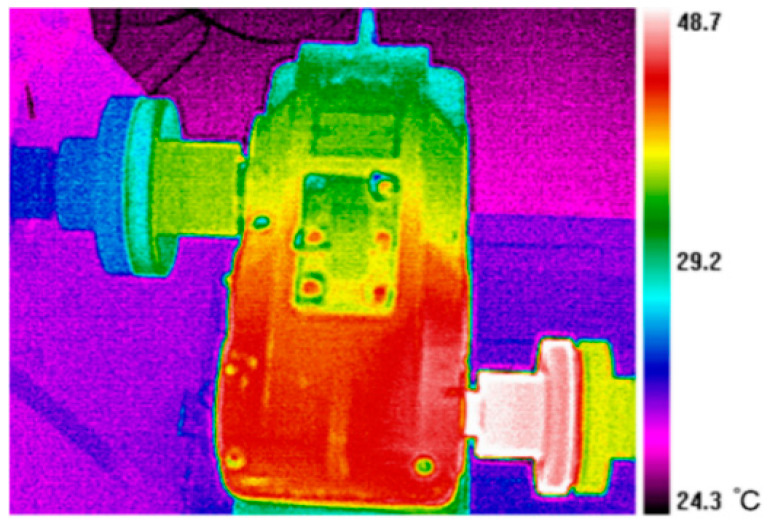
The infrared thermal image of a reducer.

**Figure 5 micromachines-13-01644-f005:**
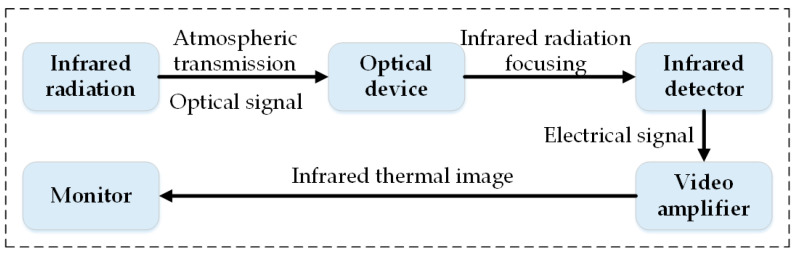
The structure and imaging principle of the infrared thermal camera.

**Figure 6 micromachines-13-01644-f006:**
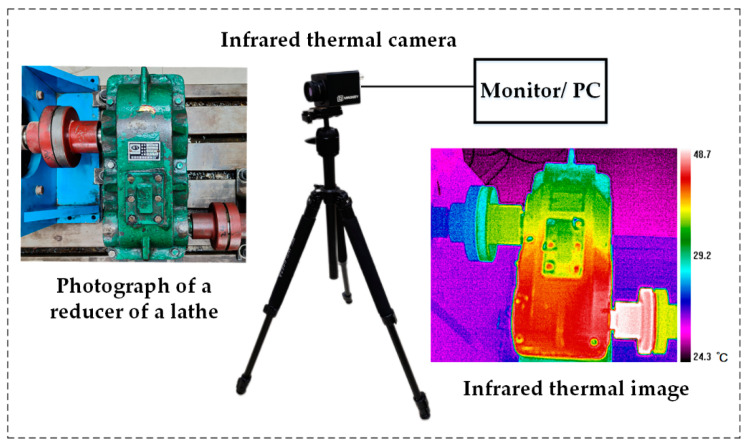
The schematic diagram of IRT experimental device.

**Figure 7 micromachines-13-01644-f007:**
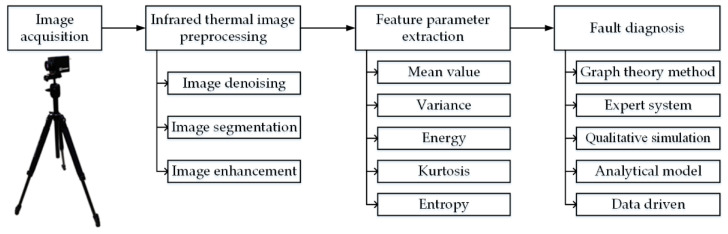
The fault diagnosis process of rotating machinery.

**Figure 8 micromachines-13-01644-f008:**
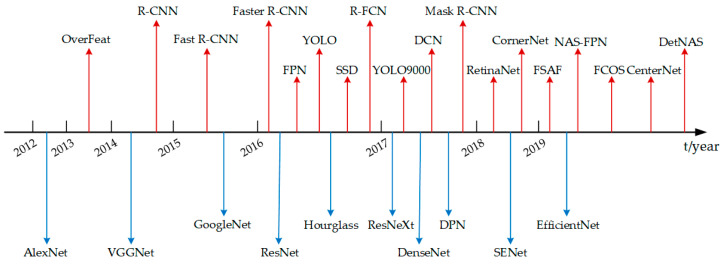
The research progress of target detection model and feature extraction networks.

**Table 1 micromachines-13-01644-t001:** The peak wavelengths of the emission spectra of typical physical events at different absolute temperatures.

Temperature (K)	Physical Significance	Peak Wavelengths (μm)
3813	The lower limit of infrared region	0.76
1811	Melting point of iron	1.60
1420	Eutectic temperature of iron–carbon alloy	2.04
1002	Eutectoid temperature of iron–carbon alloy	2.89
933	Melting point of aluminum	3.10
373	Boiling point of water	7.77
303	Room temperature	9.56
273	Temperature of ice	10.61
77	Liquefaction point of nitrogen	37.63

**Table 2 micromachines-13-01644-t002:** Summary and analysis of existing fault diagnosis methods.

Method Category	Contents	Advantage	Shortcoming
Graph theory	Symbolic directed graph [112], Fault tree [113]	The modeling is simple, the results are easy to understand and the application range is wide.	When the diagnostic object is a complex system, the search process will be very complex and the diagnostic accuracy is not high.
Expert system	Traditional expert system [114], Fuzzy expert system [115], Belief rule-based expert system [116,117]	(1) It can make use of the rich experience and knowledge of experts without mathematical modeling of the system; (2) the diagnosis results are easy to understand and widely used.	(1) It is difficult to acquire the experience and knowledge; (2) the accuracy of fault diagnosis depends on the rich experience and knowledge level of experts in the knowledge base; (3) when there are many rules, the problems, such as matching conflict and combination explosion in the reasoning process, will make the reasoning speed slow and inefficient.
Qualitative simulation	Fuzzy simulation [118], Inductive reasoning [119], Causal reasoning [120], Graph reasoning [121], Structured modeling [122]	(1) Logic is clear and easy to analyze; (2) it can infer and describe the dynamic behavior of the system.	(1) It is difficult to simulate the real environmental conditions; (2) the accumulation of a large amount of basic data is required; (3) it is difficult to establish the simulation model, and the model is not universal.
Analytical model	State estimation [123], Parameter estimation [124], Equivalent space [125]	(1) Good transparency, easy to understand and relatively simple calculation; (2) it is helpful to analyze the relationship between variables; (3) it makes use of the deep understanding of the system and has a good diagnostic effect.	(1) It is usually necessary to set various assumptions to affect the objectivity of the evaluation results; (2) there are many factors affecting the fault diagnosis, and it is difficult to construct accurate mathematical analytical formulas, with poor adaptability.
Data-driven	Information fusion [126], Multivariate statistical analysis [127], Rough set [128], Machine learning	(1) Starting from the historical data of the system, it does not need an accurate analytical model and is easier to be directly applied in the actual system; (2) it has strong self-learning and self-adaptive abilities.	(1) The information about the internal structure and mechanism of the system is less, and it is relatively difficult to analyze and explain the fault; (2) the requirements for training set and test set are high.

## Data Availability

The authors confirm that the data supporting the findings of this study are available within the article.

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
