# Peer review of "A Review of Fault Diagnosis Methods for Rotating Machinery Using Infrared Thermography"

_micromachines, 2022, doi:10.3390/mi13101644_

Round 1

Reviewer 1 Report

Reviewed article concerns review of fault diagnosis methods for rotating machinery using infrared thermography and is write in accordance with generally accepted standards of the review works. After careful reading of the submitted text there are some substantive remarks that should be taken into consideration by the Authors to improve reviewed text.

1.      The abstract should include more information about motivation to proper such review and its significance.

2.      The Authors should more carefully and more detailed describe figures (e.g., Figure 1 requires an explanation of each graphic).

3.      Consistently provide sources for each figure and table if not authored.

4.      In review articles, it is good to indicate the contribution of the authors of the text to the field by presenting the results of the author's research along with the achievements of other centers.

5.      I suggest providing the main conclusions as numbered sentences.

6.      The conclusion should be improved in term of the new knowledge gained during analysis, which should be concise with the journal scope.

Reviewer 2 Report

see attached

Round 2

Reviewer 2 Report

Accept as it is.